# Humorous peer play and social understanding in childhood
Amy Louise Paine [1] ✉, Salim Hashmi [2], Elian Fink[3], Peter Mitchell[4] & Nina Howe[5]

Humour plays a crucial role in children's early interactions, likely promoting the development of social understanding and fostering positive social relationships. To date, the connection between humour production in peer play and the development of social understanding skills in middle childhood has received limited attention. In a community sample of 130 children residing in the UK ($M = 6.16$ years old, range 5–7; 67 [51.5%] girls, 62 [47.7%] boys, and 1 [0.8%] non-binary child; 95 [73.1%] mothers and 85 [65.4%] fathers identified as Welsh, English, Scottish, or Irish), we tested our prediction that children's use of humour in play with peers would be positively associated with children's ability to understand the minds of others. We conducted detailed observational coding of children's humour production during peer play and examined associations with children's performance on a battery of social understanding assessments. Multilevel models showed that 42.8% of the variance in children's humour production was explained by play partner effects. When controlling for the effect of play partner and other individual child characteristics (age, gender, receptive vocabulary) children's spontaneous attributions of mental states were associated with humour production. Results are discussed considering how these playful exchanges reflect and influence the development of children's socio-cognitive competencies.

Humour is a central feature of a child's earliest interactions[1,2], giving critical insights into what they understand about the social world in their day-to-day interactions with others[3,4]. Throughout development, humour has important social and emotional functions: it invites interaction, enhances and maintains relationships and group cohesion, relieves tension, and enables challenging or difficult views to be expressed[5]. Importantly, humour often results in positive emotions and laughter[6] and is associated with social competence and peer acceptance[7,8]. It is proposed that children's engagement in humour is both a marker of developmental competencies and a driver of the development of socio-emotional, cognitive, and language skills[4,9,10]. Despite the importance of humour in children's peer interactions[11], it has received little research attention, and importantly, observation of humorous exchanges as peer play unfolds is rare[7,12]. Moreover, it is striking that many studies of children's behaviours during play often neglect the social nature of interaction, and recent work has indicated that a great deal of variability in a child's behaviours in play can be explained by the behaviour of their play partner[13,14]. In this study, our objective was to investigate children's humour production via observation of play with their classmates in relation to their developing social understanding skills in

middle childhood, when considering play partner influences on children's behaviour.

It is widely accepted that humour is a form of playful activity that comprises the creation or perception of incongruity; or in other words, the simultaneous occurrence of discordant elements or sudden violation of expectations, norms, or fact[5,15–18]. Much like pretend play, shared humour involves the creation and communication of alternative or distorted realities, but with the goal of eliciting amusement[10,19]. As described by Bariaud[19], "The comic hero distances himself in two ways: through fantasy and through discordance with the known world." (p. 22). Humour is also a deeply social process[17,20]. For incongruities to be perceived as humorous, rather than perplexing or frightening, they must occur within a playful frame[10,21] where ludic communicative cues are used to signal the humourist's intent, and where the audience must engage in 'emotional complicity' to share in and enjoy these intentions[19].

Children are responsive to humour from as early as three and a half months of age[2] and intentionally create humour themselves in the second half of their first year[20]. During interactions, infants create humour by 'clowning', by producing unexpected movements, facial expressions, and

[1]Cardiff University Centre for Human Developmental Science, School of Psychology, Cardiff University, Cardiff, UK. [2]Department of Psychology, Institute of Psychiatry, Psychology and Neuroscience, King's College London, London, UK. [3]School of Psychology, University of Sussex, Brighton, UK. [4]Faculty of Management, Law & Social Sciences, University of Bradford, Bradford, UK. [5]Department of Education, Concordia University, Montreal, Canada. ✉e-mail: paineal@cardiff.ac.uk

vocalisations, and by teasing others[20]. Through toddlerhood and into early childhood, children's repertoire of humorous acts grows in close connection with their motor, cognitive, and socioemotional development[10] and learning of social conventions, along with use and misuse of social rules[22,23]. In early childhood, children produce novel object-based, conceptual, and label-based humour[24]. They perform incongruities with objects (e.g., putting a teacup on their head) and produce physical incongruities (e.g., pulling funny faces) akin to clowning observed in infancy. Children also produce varied forms of verbal humour, such as playing with sound (e.g., chanting and absurd vocalisations), with language, such as deliberately mis-labelling objects or creating nonsense words and engaging in mischievous behaviour, such as teasing and play with taboo themes (e.g., bathroom humour)[24–26]. In middle childhood, children increasingly produce incongruities with double meanings (i.e., riddles and puns[11,27,28]) and engage in more complex absurd narratives and deliberate falsehoods in their play[4,11]. This increased sophistication of children's humour production emerges in parallel with more advanced linguistic skills, conceptual knowledge, and executive functioning skills, to recall, structure, and deliver punchlines with comedic timing[29,30]. Although evidence is mixed[4], boys' and girls' tendency to produce humour may diverge as children reach middle childhood, with boys becoming more frequent jokers[28].

Across development, theoretical links have been proposed between children's developing social understanding and engagement in humorous exchanges with others from infancy and into childhood[3,31–33]. Yet despite a great deal of research on different features of play in childhood, the role of humour and its connection with social development is not yet well understood[10]. Children's engagement in humorous exchanges long precedes their ability to demonstrate sophisticated social understanding skills, but sharing humorous acts – even the first years of life – are argued to indicate understanding of others' minds and expectations[3]. Of the few studies that have investigated this theorised association, most have focused on humour comprehension or appreciation rather than humour production[34]. Recent evidence identified links between parent-reports of humour and social understanding in early childhood (from infancy to 3-years-old), but not when humour was assessed in lab-based tasks administered by experimenters,[9] indicating that humour may be best understood in the context of familiar daily interactions.

Observational studies of humour production in sibling play in early and middle childhood have demonstrated links between humour and children's propensity to talk about mental states[4,26] and emotion understanding[4]; both features of children's social understanding skills[22]. These findings align with the notion that sharing humour requires sensitivity to a play partner's cognitive and emotional states: Sharing humour not only requires the joker to know that the audience will understand the truth or recognise their incongruity[32], but also to differentiate when and if their partner or audience is emotionally complicit in the play frame[5,19]. There remains, however, few observational studies that have used robust, age-appropriate assessments of developing understanding of minds[4,26]. Given that children's sharing of humour may rely on appreciating a play partner's mental states, such as belief and knowledge, and emotional cues, we investigated children's humour production in relation to age-appropriate tasks to measure children's ability to make inferences about mental and emotional states from behaviour[35–38].

However, as already noted, humour is inherently social and therefore interactional in nature. Research suggests that peer relationships become an increasingly important context for sharing humour[11], not only for pure enjoyment, to foster intimacy, maintain harmony, and diffuse potential conflicts[5,11,39]. Friendly, positive peer relationships are built upon children being 'in tune' with one another[40], and this is exemplified by recent research that highlights that children's behaviours are often heavily dependent on the behaviours of their peers in play. For example, recent findings have indicated that children's tendency to engage in pretence[13] and in connected communication[14] with a peer is dependent on their play partner's behaviour. Although the overarching focus of this paper is to examine associations between social understanding and humour production, child-level

associations could be overestimated without accounting for dyad-level effects[14]. Given that studies of focal children's play with siblings and peers have demonstrated that children engage in similar levels of humour as their play partner[4,11], we considered individual child characteristics in the context of partner effects in peer play.

The present study investigated whether humour production is associated with measures of social understanding in a sample of children on the cusp of middle childhood. Our battery of assessments included: understanding of beliefs and desires of characters in non-verbal films (Silent Films[35]), spontaneous attribution of mental states of animated geometric shapes (Triangles Theory of Mind task[36,37]), and children's ability to infer the antecedent cause of others' behaviour from their natural facial expressions (Retrodictive Mindreading[41]). Children's spontaneous humour production was coded within the context of peer play and assessed alongside parent reports. We predicted that children's performance in measures of social understanding would be associated with a greater tendency to produce humour. Given that associations between children's humour in play and social understanding could be explained by child-level characteristics, such as age, gender, language ability, and executive functioning skills, such as working memory and inhibitory control[4], we investigated these factors as potential covariates in our analyses. Finally, we used multilevel modelling to account for the dyadic context of peer play; we tested the influence of children's play partners in addition to child-level factors on humour production.

## Methods
### Participants
A community sample of 130 children residing in the UK were recruited via primary schools. Headteachers of 8 primary schools were invited to advertise the study to the families of children in Year 1 (5–6 years) and Year 2 (6–7 years). Families who expressed interest by providing their contact details were sent study information sheets and contacted to arrange a time for their online data collection session. Participating children were $M = 6.16$ years old, range 5–7 years; 67 (51.5%) were girls, 62 (47.7%) were boys, and 1 child (0.8%) was non-binary as reported by parents/caregivers. Ninety-five mothers (73.1%) and 85 (65.4%) fathers identified as Welsh, English, Scottish, or Irish. Fifty-two [40%] participating families had a gross household income of less than £29,999 (UK national average for 2022 was £32,300; Office for National Statistics, 2023) and 24 (18.5%) parent and caregiver informants were NEET (Not in Education, Employment or Training). Most children (101; 77.7%) had a sibling living in the home.

### Ethical Considerations
Ethical approval was obtained for the procedures from the Cardiff University School of Psychology Research Ethics Committee. Parents and caregivers provided consent to take part in the study and children provided verbal assent at the beginning of the online data collection session.

### Procedure
Data collection took place between June 2021 and January 2023. The main battery of child assessments, lasting approximately 1-hr, was administered on Zoom by trained research assistants to adhere to social distancing practices during the COVID-19 pandemic. Parents and caregivers (hereafter referred to as parents; 116 [89.2%] mothers, 12 [9.2%] fathers, 2 [1.5%] other caregiver) were present during the Zoom assessment sessions but were asked only to observe the child testing and not intervene unless the child became distracted and needed encouragement to engage with the session. All child assessments were presented on screen and did not require the use of a mouse. During or shortly after the Zoom child assessment, parents completed an online questionnaire. Other measures were collected during the Zoom session but are not the focus of the present study and therefore not reported. Parents and children received shopping vouchers for their time.

When Covid-19 social distancing restrictions were lifted such that researchers could visit children in-person in school ($M = 1.96$ [$SD = 1.13$] months following the Zoom assessment, range 0 – 4 months), children

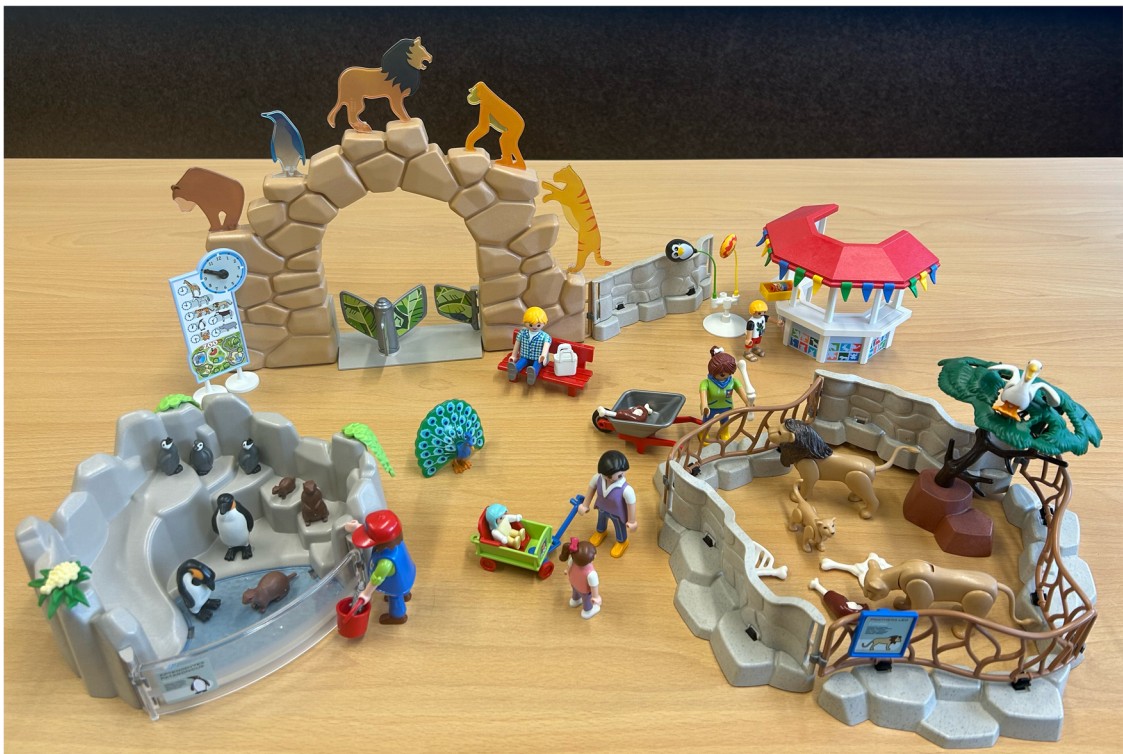

**Fig. 1 | Zoo Playmobil toy used on peer play observation.**

completed an additional small battery of child assessments 1:1 with a researcher and took part in a 15-minute play observation with another participating child from their class. Groupings were determined by teachers who advised which children would play well together. For the play observations, children were brought to a quiet area in school (i.e., wellbeing space, library) to play with a Playmobil zoo. The Playmobil zoo included animal enclosures (for penguins and lions), a zoo entrance and kiosk, a variety of zookeeper and family characters, animals (lions, penguins, birds, and meerkats), and accessories (see Fig. 1). This toy is widely used to elicit a variety of play behaviours in this age range[4,14,42]. Children were told they could play however they wanted for 15-min. Two GoPro video cameras on tripods recorded the interactions between the children at different angles, and an audio recorder was placed on the floor nearby. The researcher informed the children they would be on the other side of the door if they were needed but were left alone to play. Children received stickers and certificates for completing this phase of the study.

### Measures
**Child Humour Observed in Peer Play.** Play observation data were available for 121/130 participants in the study (3 children in one triad did not engage with the task; 2 children had changed schools by the time of the observation, 1 child was the only participating child in their class, and 1 child was not testable on the day, 2 children were audio recorded only upon parent request). Children were observed in dyads where possible (100; 82.6%), but some were observed in triads if there was an uneven number of children taking part in the study from their class (21; 17.4%). The gender composition of the groups included: 36 (29.8%) all boys, 44 (36.4%) all girls, and 41 (33.9%) mixed gender groups.

Children's humour was coded from both video recordings and transcripts of children's speech using a widely used observational coding scheme of children's humour production developed for observations of children in this age range[4,11,26,28]. Children's speech and behaviour were coded for seven categories of humour: (1) performing incongruities; (2) word play; (3) preposterous statements and humorous anecdotes; (4) sound play; (5) taboo; (6) playful teasing; and (7) clowning (see Table 1 for category

descriptions and examples). The video recordings were coded by two research assistants, who first calibrated their coding by discussing the definitions of the categories and jointly coded four play interactions with the first author. Good interrater agreement across all seven humour categorical codes was then established on an additional 22 play sessions (44/121; 36.4% children), $\kappa = 0.82$. Reliability statistics for individual categories are presented in Table 2. Categories of children's humour categories had an internal consistency of $\alpha = 0.61$ and therefore an aggregate of humour categories was created to represent total humour production. To control for slight variability in video length in play sessions (length ranged from 13.5- to 15-mins; 18 children's observations were under the 15-min target time due to bathroom breaks, interruptions, etc.), coded variables were prorated by dividing each variable by the length of the interaction to indicate rate of humour production per minute.

**Parent Reported Child Sense of Humour.** Parents completed the 23-item Children's Playfulness Scale[43,44] designed to measure five component dimensions of playfulness (physical spontaneity, social spontaneity, cognitive spontaneity, manifest joy, and sense of humour) that can be summed to construct a general playfulness factor. In this instrument ratings of a child's behaviour are made on a 5-point Likert scale ranging from (1) "doesn't sound at all like my child" to (5) "sounds exactly like my child." We harnessed the sense of humour subscale, which included 5 items (e.g., "Enjoys joking with other children") that could yield possible scores between 5 and 25. Sense of humour showed good internal consistency $\alpha = 0.73$.

### Social Cognitive Tasks
Silent Film Task. During the Zoom assessment session, children completed the Silent Films Task[35,45] to assess individual differences in theory of mind. Children watched five short film clips that depicted instances of deception, misunderstanding, and false belief from Harold Lloyd's classic silent comedy, *Safety Last![46]*. For example, one clip depicts the protagonist, Harold, sitting in the back of a van. The driver, unaware of Harold's presence, locks the van and drives away. Children watched each clip once, and after the

**Table 1 | Humour coding scheme categories and examples from the peer play observations**

| Humour Categories | Examples |
|---|---|
| **a. Performing incongruities:** Enacting a playful conflict between what is normal/expected and reality. For example, placing an object in a wrong location or making a toy perform a wrong action. | Child makes a toy lion drink from a baby bottle. Child waves a toy tiger in the air saying, "The tiger's flying!" |
| **b. Word play:** Nonsense words, rhyming words, riddles, jokes, label-based humour. Making deliberate mistakes in language or changing words in well-known songs. | "Let's go to the zoo, zoo, zookini!" "And the penguits go here." |
| **c. Preposterous statements and humorous anecdotes:** Creating absurd or unusual stories, anecdotes, or making announcements, nonsense sentences, deliberate falsehoods (identified by conflicting statements). | "What if the peacock just does this… chases him!" "Your son's gonna get eaten by a peacock!" |
| **d. Sound play:** Over exaggerated vocalisations or speech, exaggerated gasps, animal noises, using a very deep or gruff voice in a silly or unconventional way (e.g., fast or slow), or using silly accents, chanting, bursting into exaggerated song. | Child singing, "Oh my god, there's so much meat meat meat meat meat!" In a silly accent, "Give me the babeh!" |
| **e. Taboo:** Disgusting noises, such as blowing raspberries, fart noises, burp noises, using taboo words or discussion and/or enacting taboo themes. Includes violent themes of play, like stabbing, shooting, or terms like "die!" Any play that is rule breaking (yet playful) in nature. | "[Singsong] Bye bye lion… and he killed the lion!" "An ape is an evil type of monkey that kills you." |
| **f. Playful teasing:** Light-hearted, playful, mischievous behavior directed to play partner. Includes light-hearted insults and playful rough and tumble. Must be coupled with playful cues (smiling, laughter, playful tone of voice). | "You missed! [Peer Name]-o-oodle!" "You know, I'll tell – 'cause you're talking too loud!" |
| **g. Clowning:** Silly or over exaggerated body movements, dancing, posing, or pulling funny faces. | Child falls dramatically to the floor. Child sticks out tongue and pulls faces. |

*Note.* Categories of humour could co-occur.

**Table 2 | Means, standard deviations, ranges of rate (per minute of play) and intra-class correlations (ICC) of prorated raw scores for total humour, humour types, and parent-reported humour, in addition to the percentage of children engaging in each behaviour at least once**

| | Mean (SD) | Range | % | *ICC* | κ | Girls = 64 | Boys = 56 | *t* test statistic, *p value*, Mdiff *(SE)*, 95% CI | Cohen's d |
|---|---|---|---|---|---|---|---|---|---|
| Total humour | 1.81 (1.10) | 0–4.93 | 99.17 | 0.43, *p* < 0.001 | 0.82 | 1.61 (0.97) | 2.03 (1.21) | *t*(118) = 1.68, *p* = 0.10, 0.13 (0.08), 0.03 to 0.29 | 0.31 |
| Performing incongruities | 0.13 (0.17) | 0–0.80 | 64.46 | 0.12, *p* = 0.26 | 0.76 | 0.10 (0.14) | 0.16 (0.19) | *t*(118) = 2.07, *p* = 0.04, 0.09 (0.04), 0.003 to 0.18 | 0.38 |
| Word play | 0.05 (0.77) | 0–0.33 | 43.80 | 0.25, *p* = 0.04 | 0.93 | 0.06 (0.08) | 0.04 (0.07) | *t*(118) = −1.15, *p* = 0.25, −0.04 (0.03), −0.10 to 0.03 | 0.21 |
| Preposterous statements and humorous anecdotes | 0.59 (0.48) | 0–2.20 | 95.87 | 0.23, *p* = 0.08 | 0.84 | 0.48 (0.37) | 0.72 (0.57) | *t*(93.87) = 1.87, *p* = 0.07, 0.11 (0.06), −0.01 to 0.23 | 0.35 |
| Sound play | 1.08 (0.81) | 0–3.73 | 97.52 | 0.50, *p* < 0.001 | 0.95 | 1.01 (0.81) | 1.15 (0.81) | *t*(118) = 0.78, *p* = 0.44, 0.06 (0.08), −0.09 to 0.21 | 0.14 |
| Taboo | 0.24 (0.24) | 0–1.13 | 81.81 | 0.16, *p* = 0.22 | 0.85 | 0.17 (0.20) | 0.31 (0.26) | *t*(118) = 2.99, *p* = 0.003, 0.14 (0.05), 0.05 to 0.23 | 0.55 |
| Playful teasing | 0.15 (0.20) | 0–0.87 | 61.98 | 0.28, *p* = 0.03 | 0.92 | 0.14 (0.18) | 0.17 (0.22) | *t*(118) = 0.78, *p* = 0.44, 0.04 (0.05), −0.06 to 0.14 | 0.14 |
| Clowning | 0.06 (0.10) | 0–0.53 | 33.88 | 0.43, *p* < 0.001 | 0.64 | 0.04 (0.09) | 0.07 (0.12) | *t*(106.55) = 1.42, *p* = 0.16, 0.05 (0.04), −0.02 to 0.12 | 0.26 |
| Parent rated sense of humour | 19.65 (3.56) | 7–25.00 | - | - | - | 18.82 (2.97) | 20.57 (3.96) | *t*(116) = 2.74, *p* = 0.007, 1.75 (0.64), 0.49 to 3.02 | 0.51 |

*Note. N* = 121. *ICC* Intraclass Correlation. Note that boys and girls = 120 as one child identified as non-binary and was therefore excluded from this analysis. Descriptive data of observed humour production represents prorated raw scores of humour production (rate per minute). *t*-test analyses for observed humour production are based on SQRT transformed data.

clip were asked questions about each video, such as, "Why did the driver lock Harold in the van?" Children's responses to the six questions asked in the task were transcribed and coded by research assistants who were unaware of the goals of the study. The transcripts were coded using Devine and Hughes' (2013) protocol. A correct answer with mention of an explicit mental state (e.g., 'believe', 'think', 'know') was coded as 2 (fully correct). Answers that referred to the facts, outcome, trait or motivation without mention of an explicit mental state were coded as 1 (partially correct), and factually incorrect answers received scores of 0 (incorrect). As such scores on this task could range from yielding a possible score for each child between 0 and 12. An independent coder coded 30 (23.6%) to establish interrater reliability; kappa values were within acceptable ranges (κ = 0.76 to 1.00) for each item. Internal consistency of final coded clips was α = 0.52.

Triangles Task. The Triangles Task[36,37] was administered during the Zoom assessment to assess mental-state attribution. Children viewed five short

clips depicting a large red triangle and a small blue triangle moving about on a white background, which were designed to imply instances of encouraging, teasing, surprising, (theory of mind animations), fighting, and following (goal-directed animations). After each video, children were asked "What happened in the cartoon?" No feedback was given to children's responses, except general comments to support the child's engagement with the task. The animations were presented in two counterbalanced orders. Children's responses were first coded for the degree of mentalising in their descriptions of the cartoons, ranging from 0–3 for each video. No response/I don't know was coded as (0). Any response comprising a simple action with no mention of interaction or of mental states was coded as action (1). Responses that explicitly mentioned interaction between the triangles was coded as interaction (2). Finally, any descriptions that included explicit mention of mental state terms (e.g., *think, know, want, feel*) were coded as mentalising (3). Two independent raters coded 33 (26.2%) children's responses to the Triangles task; kappa values were within acceptable ranges

(κ = 0.74–0.85 for the individual cartoons). The final variable used for analyses was the number of theory of mind animations for which children provided mentalising descriptions (ranging from 0–3). Internal consistency of coded descriptions was α = 0.66.

Retrodictive Mindreading Task. Retrodictive mindreading tasks are designed to assess children's ability to discriminate between dynamic, subtly expressed natural emotional displays and behaviour[38,41]. The task assesses whether a child can accurately interpret another person's inner state but also infer the event that caused that inner state. This task was administered on Zoom and was an adapted version of previous retrodictive mindreading tasks[41,47]. Children viewed videos of people's (target's) reactions as they were 'getting a present' and 'being told off'. Children were presented with each target's videos simultaneously on the screen and were asked to look at them both carefully and decide which video was the one where the target was 'getting a present'. As a practice, children were shown videos for two targets to instruct them how to respond (indicate video on the left or right; children's responses were confirmed by the experimenter to ensure they could label left or right accurately. In cases where children found this challenging, they could point to the screen and parents would indicate their choice to the researcher. Children then went on to make judgements on 20 targets in the task itself. Children's accuracy scores in identifying the target getting the present could range from 0 to 20. Internal consistency was α = 0.61.

### Child Covariates

Receptive Vocabulary. Children's vocabulary knowledge was assessed using the British Picture Vocabulary Scale (BPVS-III[48,49]) during the Zoom assessment. The BPVS is a well-established, valid and reliable measure of receptive language ability[49]. In this task, the researcher spoke a word to the child, who was asked to identify the picture that corresponded to the word from four possible response options. Children received two practice trials where feedback was given if incorrect. The task was terminated after children exceeded a predefined threshold of errors.

Working Memory. The Picture Sequence Memory Test (PSMT) from the NIH Toolbox[50] was selected to assess episodic working memory and was administered during the in-person school assessment. The NIH toolbox has been validated against other neuropsychological assessments and demonstrated excellent psychometric properties[51,52]. Children were presented with a series of pictures on a computer tablet (from 6–18 pictures depending on age) depicting activities accompanied by audio descriptions (e.g., "Fly a kite" or "Play in the sand"). After each series of pictures, children were presented with all pictures from the sequence and were asked to drag and drop the images into the correct order. Children's scores were based on the cumulative number of adjacent pairs placed correctly over two trials.

Inhibitory Control. Children were administered the "Flanker" Inhibitory Control and Attention Test from the NIH Toolbox[50] during the in-person school assessment. In this assessment, children were required to match to a target stimulus while inhibiting attention to its flanking stimuli (fish) presented on a computer tablet. In some trials, the target stimuli pointed in the same direction as the flanking stimuli (congruent trials), and in others the target stimuli pointed in the opposite direction (incongruent trials). Task instructions were given verbally by the experimenter with accompanying practice trials. Children were then presented with 20 fish test trials; if they scored ≥ 90% they were given 20 additional trials with arrow targets. Children's inhibitory control scores were based on their performance on both the congruent and incongruent trials.

## Results
### Plan of analysis
We first present descriptive statistics of children's humour production in play, differences by child gender and the gender composition and size of the group that children were observed within during peer play (i.e., dyads vs triads). For the purpose of description, we present proportioned raw scores (rate per minute, to account for small differences in play observation length) of children's humour production. Our predictions regarding associations between children's total humour production and social understanding (amongst other predictions not addressed in this paper) were pre-registered https://osf.io/6cpvw (date of pre-registration 10/12/2022). For exploratory purposes we also describe and present basic analyses for subtypes of humour that formed the total humour production variable. As is typical with coded play data, children's humour data were skewed; therefore, all scores were SQRT transformed prior to analyses, which helped to normalise the distribution. We tested differences in children's humour production according to child gender and gender composition of the peer group in the observed play session; for these analyses, one non-binary child was excluded from the analysis for child gender due to low cell size but included in analyses pertaining to gender composition of the peer group. Associations amongst humour codes as well as bivariate associations across humour codes and child factors (age, receptive vocabulary, executive functioning, and social understanding variables). As recommended by Kenny, et al.[53], we used Spearman's correlations as a more conservative approach given that children were nested within pairs or triads for the play observation. This enables comparisons across different studies in this literature that use this approach with dyadic data; however, these findings must be interpreted cautiously. Zero-order correlations between social understanding measures and total humour production, $p < 0.05$ were followed up in subsequent analyses controlling for identified covariates.

We adopted a multi-level modelling approach (MLM) to test direct associations between child factors and total humour production observed in the play interaction. This deviated from our pre-registered analysis plan, but we took this approach for several reasons. First, MLM enables us not just to account for, but moreover to evaluate the extent to which children's humour production is explained by partner effects[53]. The importance of this is underscored by recent papers indicating the overwhelming partner effects on children's observed play behaviours[13,14]. Second, MLM allows both child-level variables and group-level variables to be modelled. In our study, children were observed in dyads and triads. MLM provides the advantage of accounting for this group-level variable in our models that would not have been possible otherwise.

We conducted a baseline model (Step 0), to estimate the variance in children's total humour production that could be explained by similarities in how children played within the context of their dyads and triads. Next, we included predictors as fixed effects (Step 1). All child-level continuous predictors were grand mean centred. In all analyses, group size was controlled as a group level (Level 2) predictor. We tested incremental model fit from Step 0 to Step 1 to evaluate if the child- and group-level predictors explained more variability in humour production beyond the effect of the play partners. This was determined by comparing log-likelihood ratios. Regression models were conducted using MPlus version 8[54]. Missing data were accounted for using maximum likelihood with robust standard errors (MLR) as an estimator. To assess incremental model fit, the chi-square difference test was based on log-likelihood ratios and scaling correction factors obtained with the MLR estimator[55].

### Descriptive statistics
Descriptive statistics for children's humour during peer play are presented in Table 2. All children except one produced at least one instance of humour in the peer play session. For exploratory purposes, categories of children's humour are described. Sound play was the most common humour category produced, characterised by over-exaggerated vocalisations, silly accents, or rhythmic speech, for example, "Oh my god, there's so much meat meat meat meat meat". Coded categories often co-occurred, as exemplified by this instance of playful teasing and word play, "You missed! [Peer Name]-o-oodle!" Intraclass correlation coefficients indicated significant partner effects in children's total humour production, $ICC(57) = 0.43$, 95% $CI(0.23$ to $0.62)$ $p < 0.001$. This was also the case for subcategories of humour, including word play, sound play, playful teasing, and clowning (see Table 2).

There were no statistically significant differences in total humour production produced per minute between children who were observed in dyads ($M = 1.77$, $SD = 1.07$) and those in triads ($M = 1.97$, $SD = 1.26$), $t(119) = -0.64$, $p = 0.53$, nor were there any statistically significant differences according to whether children were observed in a same ($M = 1.66$, $SD = 1.09$) or a mixed ($M = 1.88$, $SD = 1.13$) gender dyad or triad, $t(119) = -0.57$, $p = 0.57$. There were, however, statistically significant differences by child gender, where boys produced more incongruities and taboo humour compared to girls. Significant differences were also found in parent reports, where parents similarly reported higher sense of humour scores for boys compared to girls (see Table 2). There was no statistically significant evidence of gender differences detected for any other child variables of interest: receptive vocabulary, $t(115) = 0.70$, $p = 0.49$, inhibitory control, $t(117) = -0.77$, $p = 0.44$, working memory, $t(118) = 1.03$, $p = 0.31$, and social understanding task scores including the Triangles task, $t(116) = 0.46$, $p = 0.64$, Silent Films, $t(116) = -1.09$, $p = 0.28$, and Retrodictive Mindreading, $t(116) = -0.56$, $p = 0.58$.

### Child factors associated with children's humour

Bivariate correlations between all variables of interest are presented in Table 3. Firstly, it is noteworthy that children's total spontaneous humour production in peer play was significantly and positively associated with parent reports of their child's sense of humour, $r_s(117) = 0.21$, $95\% CI(0.03\ to\ 0.38)$, $p = 0.02$. We did not detect evidence of a significant association between children's total observed humour (observed or parent reported) and child age, working memory, inhibitory control, or receptive vocabulary.

### Social understanding and humour

There was no statistically significant evidence of an association between parent reports of children's sense of humour and children's performance on the measures of social understanding. For children's performance on the social understanding tasks, it is noteworthy but unsurprising that correlations between tasks were non-significant or small given that social understanding is argued to comprise distinct abilities[56]. Children's total observed humour was significantly positively correlated with children's performance on the Triangles task, $r_s(115) = 0.20$, $95\% CI(0.01\ to\ 0.37)$, $p = 0.03$ (but not social understanding as assessed by the Silent Films or Retrodictive Mindreading tasks). There were also significant positive associations detected between children's performance on the measures of social understanding and subtypes of humour production: between the Silent Films task and playful teasing, $r_s(117) = 0.35$, $95\% CI(0.17\ to\ 0.50)$, $p < 0.001$, between the Retrodictive Mindreading task and clowning humour, $r_s(117) = 0.20$, $95\% CI(0.01\ to\ 0.37)$, $p = 0.03$, and the Triangles task and both preposterous statements, $r_s(115) = 0.19$, $95\% CI(0.003\ to\ 0.36)$, $p = 0.04$, and sound play, $r_s(115) = 0.22$, $95\% CI(0.03\ to\ 0.39)$, $p = 0.02$.

In accordance with our pre-registered hypothesis, we detected a significant association between children's performance on the Triangles task and their total humour production in peer play. Therefore, we used multilevel models to evaluate the amount of variability in children's humorous play that could be explained by similarity in behaviours amongst children observed within the same group, in addition to the role of child level predictors. Child age, gender, and receptive vocabulary were included in this analysis as these variables were associated children's total humour production or children's performance on the Triangles task (Table 4).

For total humour production, the baseline model indicated that 42.8% of the variability in children's humour production was explained by dependencies amongst children in dyads and triads. We controlled for group size as a group level (Level 2) predictor, and child-level (Level 1) variables of age, gender, receptive vocabulary and children's performance on the Triangles task. Addition of these predictors significantly improved model fit, TRd = 15.67, $df = 5$, $p < 0.05$, explaining an additional 2.7% of the variance in humour produced in play. At this step, notably only children's social understanding in the Triangles task predicted children's humour, $est. = 0.27$ ($SE = 0.11$), $p = 0.01$.

## Discussion

Humour is a prominent feature of children's play and is considered one of the building blocks of positive social relationships[26]. Although being able to conceive and express humour has long been thought to be closely related to children's capacity to understand the mental states and feelings of others[3,32,33], few studies have investigated this theorised association. Moreover, much of the extant research has overlooked the social nature of play and the influence of play partners on children's behaviour[13,14]. In this study we observed children's humour production during peer play and investigated associations with a battery of social understanding tasks, while accounting for potential child-level covariates and dyadic influences during the interaction. Accordingly, the present study contributes to understanding of the nature and underpinnings of humour production in middle childhood.

As in previous studies of peer play[11] and across different contexts[4,11,28], most children produced humour at least once during play with a peer. Children most often produced sound play, which often co-occurs with other forms of humour as a ludic cue to signal playful intentions[5,19,23] (e.g., vocalising, singing, chanting, or using a silly accent). It was also common for children to produce linguistically complex forms of humour, such as telling preposterous stories in their play (e.g., "Your son's gonna get eaten by a peacock!"). We observed no significant differences between boys' and girls' total use of humour, although boys were more likely to perform incongruities with objects (e.g., making a toy lion drink from a baby bottle) and engage in taboo humour, involving disgusting or violent themes (e.g., "An ape is an evil type of monkey that kills you."). These gender differences were also evident in parent reports of children's sense of humour, in line with previous studies[28,57].

We investigated associations between children's social understanding and spontaneous humour production during peer play. Consistent with our hypothesis, children who engaged in spontaneous attribution of mental states to animations in the Triangles task[37] were more likely to produce humour, even when play partner effects and other child-level covariates were controlled. Although individual differences effects were small relative to the influence of partner effects, our findings contribute to a limited but growing body of evidence that observing children's humour provides a window to understanding their developing social understanding competencies[4,26]. However, the cross-sectional nature of the present study precludes us from making any assertions about causal relationships between social understanding and children's humour production. Therefore, it will be useful in future research to disentangle predictive relationships longitudinally. As indicated by parent reports of humour and social cognition in early childhood, early humour predicted social understanding 6 months later, but not the reverse[9]. This may suggest that engaging in humour may be a positive and safe context for young children to develop understanding of others' intentions, desires, and beliefs, and hone their social understanding skills[3]. Future studies should consider whether similar directional associations can be observed in middle childhood.

Although we did not detect associations between performance in all social cognitive tasks and total humour production, it is noteworthy in our exploration of subtypes of children's humour that playful teasing (i.e., light-hearted mischievous behaviour with playful cues) was associated with children's understanding of beliefs and desires, as demonstrated in the Silent Films task. Engaging in playful teasing exchanges with others may be a particularly important experience where children must harness their knowledge of others' mental states, given risk of causing offense and upset if not understood or appreciated by a play partner[26]. This is particularly pertinent in peer relationships, which are voluntary and characterised by mutual attachment, liking, and reciprocity, and could potentially be jeopardised should teasing be upsetting[58].

The current study also demonstrated a strong play partner effect on children's humour production. Consistent with recent studies of children's engagement in pretence and connectedness[13,14], the effect of play partners accounted for a moderate proportion of the variance explained in children's humour production (over 40%). This finding has critical

**Table 3 | Spearman bivariate correlations between variables of interest**

| | 1. | 2. | 3. | 4. | 5. | 6. | 7. | 8. | 9. | 10. | 11. | 12. | 13. | 14. | 15. | 16. |
|---|---|---|---|---|---|---|---|---|---|---|---|---|---|---|---|---|
| 1. Total humour | - | | | | | | | | | | | | | | | |
| 2. Performing incongruities | 0.48** | - | | | | | | | | | | | | | | |
| 3. Word play | 0.22* | 0.14 | - | | | | | | | | | | | | | |
| 4. Preposterous statements and humorous anecdotes | 0.82** | 0.53** | 0.06 | - | | | | | | | | | | | | |
| 5. Sound play | 0.90** | 0.33** | 0.18* | 0.65** | - | | | | | | | | | | | |
| 6. Taboo | 0.57** | 0.42** | 0.07 | 0.51** | 0.36** | - | | | | | | | | | | |
| 7. Playful teasing | 0.36** | 0.04 | -0.08 | 0.30** | 0.15 | 0.23* | - | | | | | | | | | |
| 8. Clowning | 0.31** | 0.27** | 0.11 | 0.17 | 0.20** | 0.33** | 0.16 | - | | | | | | | | |
| 9. Child age | 0.07 | 0.05 | -0.17 | 0.07 | 0.01 | 0.04 | 0.16 | 0.25** | - | | | | | | | |
| 10. Parent reported sense of humour | 0.21* | 0.13 | -0.06 | 0.10 | 0.18 | 0.20* | 0.26** | 0.21* | 0.16 | - | | | | | | |
| 11. Receptive vocabulary | 0.13 | 0.16 | -0.13 | 0.17 | 0.09 | 0.05 | 0.17 | 0.28** | 0.47** | 0.04 | - | | | | | |
| 12. Inhibitory control | -0.12 | -0.07 | -0.14 | -0.11 | -0.09 | -0.10 | 0.02 | 0.03 | 0.06 | -0.10 | 0.25** | - | | | | |
| 13. Working memory | -0.03 | -0.13 | -0.09 | -0.02 | 0.01 | -0.04 | 0.05 | -0.14 | -0.06 | -0.10 | 0.13 | 0.23** | - | | | |
| 14. Triangles ToM | 0.20* | 0.15 | -0.10 | 0.19* | 0.22* | 0.12 | -0.08 | 0.16 | 0.27** | 0.07 | 0.24** | 0.06 | 0.08 | - | | |
| 15. Silent Films | 0.05 | -0.09 | -0.14 | 0.07 | 0.07 | -0.17 | 0.35** | 0.02 | 0.27** | 0.07 | 0.25** | 0.21** | 0.13 | 0.14 | - | |
| 16. Retrodictive Mindreading | 0.09 | 0.10 | 0.03 | 0.15 | 0.07 | 0.09 | -0.04 | 0.20* | 0.27** | 0.03 | 0.33** | 0.07 | 0.02 | 0.20* | 0.24** | - |
| Available n | 121 | 121 | 121 | 121 | 121 | 121 | 121 | 121 | 121 | 119 | 118 | 120 | 121 | 117 | 119 | 119 |
| M (SD) | 1.27 (0.43) | 0.27 (0.24) | 0.15 (0.18) | 0.70 (0.33) | 0.96 (0.41) | 0.41 (0.27) | 0.28 (0.27) | 0.13 (0.20) | 81.99 (7.52) | 19.65 (3.56) | 90.92 (17.52) | 35.78 (7.87) | 8.23 (4.27) | 0.33 (0.56) | 2.64 (2.34) | 14.05 (2.94) |
| Range | 0–2.22 | 0–0.89 | 0–0.58 | 0–1.48 | 0–1.93 | 0–1.06 | 0–0.93 | 0–0.73 | 63–97 | 7–25 | 45–133 | 14–40 | 1–26 | 0–2 | 0–9 | 4–19 |

Note. ToM Theory of Mind. **$p < 0.01$, *$p < 0.05$. N ranges from 114 to 121 of data available out of 121 children in the present sample with observational play data. Descriptive data for coded humour data represents SQRT transformations used for analysis. Child age represents age in months at the time of the peer play observation.

**Table 4 | Multilevel regression model examining the association between child-level factors and children's total humour production in play**

| | Parameter estimates | Estimate (SE) | 95% *CI* | *p* value |
|---|---|---|---|---|
| Step 0 | *Fixed effects* | | | |
| | Child-level | | | |
| | Log-likelihood | −62.56 | | |
| | Partial *ICC* | 0.428 | | |
| Step 1 | *Fixed effects* | | | |
| | Child-level (Level 1) | | | |
| | Age | −0.04 (0.15) | −0.34 to 0.26 | 0.80 |
| | Gender | −0.09 (0.11) | −0.29 to 0.12 | 0.41 |
| | Receptive vocabulary | −0.02 (0.13) | −0.27 to 0.23 | 0.87 |
| | Triangles ToM | 0.27 (0.11) | 0.06 to 0.48 | 0.01 |
| | Group-level (Level 2) | | | |
| | Group size | 0.07 (0.16) | −0.25 to 0.39 | 0.66 |
| | Intercept | 4.40 (0.97) | 2.49 to 6.30 | < 0.001 |
| | Log-likelihood | −54.42 | | |
| | Partial *ICC* | 0.401 | | |

*Note. ToM* Theory of Mind. Group size is controlled as a Level 2 (group-level) factor. Final model *N* = 113.

theoretical and methodological implications. Developmental theories of humour have emphasised the cognitive components of children's perception and understanding of incongruities[16,18], but this research underscores more recent assertions that humour is fundamentally social[3]. Not only is humour so often overlooked as a feature of play behaviours, but when studied in childhood, it is rarely investigated observationally within their naturalistic interactions[7,12]. More research is needed to understand the extent of social influence across different relationships, as evidence suggests children's humour may become increasingly specific to their play partner as they develop close relationships outside of the family[11]. Vital next steps include observing children's humour production across different social contexts, as a function of relationship quality or the social skills of their play partner, and how children and their play partners produce and respond to humorous acts. Previous research indicates that children who produce humour that is imitated by their play partner also talk more about cognitive states, which may be indicative of their ability to build connected sequences, but eliciting negative responses is associated with less positive rapport between children[26]. Although beyond the scope of the present study, how children respond to and extend one another's humorous acts in play sequences would be a worthy avenue for future research. Our research also highlights that future observational studies of children's play must account for partner effects in statistical models[13].

## Limitations

Although this study harnessed data from parent reports, child observations, and a battery of child assessments in a moderately-sized sample, there are some limitations to note. First, to adhere to social distancing restrictions during the COVID-19 pandemic, child assessments were conducted online rather than in person, which has benefits as well as limitations. Conducting assessments online in the family home is environmentally friendly, less resource-intensive, and more convenient for families who may otherwise not be involved in research[59]. Online approaches also decrease some data collection challenges, such as audio and visual problems for coding data, but distractions (e.g., child leaving the room, siblings interfering) and maintaining child engagement is more challenging[60]. Second, we must acknowledge that – as is common with advanced tasks to assess understanding of mental states[61], internal consistencies of social understanding tasks varied. Particularly, the Silent Films task demonstrated unacceptably low internal consistency; it is quite possible that this task was too challenging

for this age group. Third, as a common trade-off when collecting rich observational data within a sufficiently powered sample size[13,14,42], our observations of children's humour production were based on relatively short play sessions in a single context. Although we noted a small but significant association between children's observed humour production and parent reports of their child's humour, it is quite possible that the nature of children's humour production may vary across different durations or contexts of play[28]. For example, although children were left to play privately, their engagement in more rambunctious behaviours like clowning, taboo humour, and playful teasing may have been inhibited by the school context or by simply knowing they were being video recorded.

Given the influence of play partner effects in this study, it is important to note that children were grouped for observation according to teacher recommendations of play partners, however, it is likely that relationship quality varied across groups of children. Given that children are more likely to engage in humour when they have positive rapport with others[26], future studies should include an index of relationship quality to further investigate the role of play partner for individual children's humour production. Finally, additional child characteristics may play an important role in children's production of humour and should be considered as covariates in future studies. Although receptive vocabulary was associated with children's performance in measures of social understanding, there was no evidence of an association with humour production. Expressive language competence, but not receptive language, is associated with behaviours in peer play, such as assigning roles for pretence[13]. It would seem likely that children who can use language effectively may be better able to engage in complex forms of humour, such as telling preposterous stories or producing conceptual incongruities in word play.

## Conclusion

Sharing humour is complex in its integration of emotional, social, and cognitive processes[3]. Social interaction is the primary context for engagement in humour, yet how children spontaneously produce humour in their play with others has largely been overlooked, thus prompting our more detailed examination. Our findings support previous evidence that humour provides important insights into children's developing social cognitive skills. Furthermore, the notable influence of play partner on humour production underscores humour as a social phenomenon in childhood, and highlights the need for further research considering how these interactions unfold within and across different social relationships across development.

## Data availability

Raw data for the present study for participants who consented to data sharing are available on the ReShare UK Data Archive via Paine, Amy (2025). *Playful Minds: Humorous Peer Play and Social Understanding in Childhood, 2021–2023*. [Data Collection]. Colchester, Essex: UK Data Service. https://doi.org/10.5255/UKDA-SN-857661. Data are embargoed until published.

## Code availability

Analyses were conducted using SPSS version 27 and MPlus Version 8. MPlus code for analyses are available via Paine, A. L. (2025). Humour in Peer Play and Social Understanding. Retrieved from osf.io/d5bnh.

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

## Acknowledgements

The Child HUMour Study (CHUMS) was funded by the Economic and Social Research Council (grant reference ES/T00049X/1). The funders had no role in study design, data collection and analysis, decision to publish or preparation of the manuscript. We are grateful to the participants who took part in this study and to Charlotte Robinson, Rhys Davies, Benny Wu, Molly Sandilands, and Amy Walker for their research assistance.

## Author contributions

Amy Paine: Conceptualization, Funding Acquisition, Project Administration, Investigation, Methodology, Data Curation, Supervision, Formal Analysis, Writing – Original Draft Preparation, Writing – Review & Editing. Salim Hashmi: Investigation, Data Curation, Supervision, Writing – Review & Editing. Elian Fink: Formal Analysis, Writing – Review & Editing. Peter Mitchell: Conceptualization, Funding Acquisition, Methodology, Writing – Review & Editing. Nina Howe: Conceptualization, Funding Acquisition, Methodology, Writing – Review & Editing.

## Competing interests

The authors declare no competing interests.
