## [Transparent Peer Review file · Communications Psychology]

Humorous Peer Play and Social Understanding in Childhood

Corresponding Author: Dr Amy Paine

Version 0:

Decision Letter:

Dear Dr Paine,

Thank you for your patience during the peer-review process. Your manuscript titled "Playful Minds: Humorous Peer Play and Social Understanding in Childhood" has now been seen by 2 reviewers, and I include their comments at the end of this message. They find your work of interest but raised some important points. We are interested in the possibility of publishing your study in Communications Psychology, but would like to consider your responses to these concerns and assess a revised manuscript before we make a final decision on publication.

We therefore invite you to revise and resubmit your manuscript, along with a point-by-point response to the reviewers. Please highlight all changes in the manuscript text file.

Editorially, we consider it important that the revised manuscript address the conceptual and methodological concerns of the reviewers. We request that you provide additional analysis of partner effects as requested by Reviewer 2.

Editorially, we welcome the existence of preregistration, and ask you to ensure that your revision complies with our respective guidelines to facilitate future steps. We ask authors to be precise with regard to whether only hypotheses, or hypotheses and analyses were preregistered. All originally preregistered hypotheses and analyses should be included, unless scientifically unsound, in which case the deviation needs to be highlighted and explained. Additional (exploratory) analyses may be included, but need to be labelled as post-hoc, non-preregistered. The full policy is here: <https://www.nature.com/commspsychol/editorial-policies/preregistration-policy>.

Please ensure you follow our statistical guidelines when reporting statistics (<https://www.nature.com/commspsychol/submit/submission-guidelines#statistical-guidelines>). Please note in particular our requirements for the reporting and interpretation of null-results. Non-significant findings derived from null-hypotheses significance tests should be reported in full, but may not be interpreted. Where you interpret null results, this interpretation must be based on Bayes Factors or equivalence tests.

I am attaching an Editorial Requests Table that details critical reporting requirements for the revised manuscript. Please attend to each item and ensure your manuscript is fully compliant. If your revised manuscript is not aligned with these requests on major issues, such as those concerning statistics, it may be returned to you for further revisions without re-review.

Please submit the following items:

- Revised manuscript
- Point-by-point response to the referees' comments

- Cover letter (as a separate document)
- [Nature Research Reporting Summary](https://www.nature.com/documents/nr-reporting-summary.zip)
- [Editorial Policy Checklist](https://www.nature.com/documents/nr-editorial-policy-checklist.pdf)
- Completed Editorial Request Table (attached).

via this link: Link Redacted .

Additional guidance is available in our style and formatting guide [Communications Psychology formatting guide](https://www.nature.com/documents/commpsychol-style-formatting-guide-accept.pdf).

Best regards,

Jennifer Bellingtier

Jennifer Bellingtier, PhD
Senior Editor
Communications Psychology

REVIEWER EXPERTISE:

Reviewer #1 Social-cognitive development
Reviewer #2 Social-cognitive development

REVIEWER REPORTS:

Reviewer #1 (Remarks to the Author):

The paper presents a novel analyses containing multiple data sources concerning children's humor in middle childhood. It can contribute to advance the field by combining multiple methods that include both tests, and naturalistic data from children's peer group interactions. The results can therefore influence thinking in the field and contribute to the understanding of both children's humor production, and comprehension.

It is an original study that presents new analysis and knowledge. The uniqueness of the study lies in its coding of naturally occurring (albeit short) children's peer group interactions and humorous situations occurring there. The authors demonstrate knowledge and sensitivity to the contextual and social relational character of humor (humor as dependent on the situational context, social constellations and the 'common ground' (see Clark, 1991) between the participants), and the cognitive aspects (incongruency etc.) of humor production and comprehension.

The data and analysis are technically sound. The variety of tests is relevant to cover various cognitive and social/normative aspects related to humor production/ comprehension. They are highly relevant to answer research questions. However, the argumentation concerning why and how these tests were selected needs to be introduced early on, e.g., by relating them to the review of prior research and to research questions. Conclusions of the paper are based on sound analytical evidence, and is well-suited to the field of psychology, and to other fields related to child studies. The paper deals with ethics in an informed and relevant manner.

The study is relevant to the journal. There are some issues however that needs attention. There needs to be a better and convincing presentation of how the different tests were selected, and how they relate to the cognitive and social abilities relevant for the research question. Data was collected during the Covid-19 pandemics. It would be relevant to know the

authors' discussion of how zoom based data collection can affect the quality and characteristics of data? How were the participants selected? The authors can provide some review on the different kind of humor, and its relation to children's age (e.g., analysed in naturalistic interactions). In all, I think that the study is novel, provides interesting and novel results, and can be published (pending on the results of revisions).

Reviewer #2 (Remarks to the Author):

Communications Psychology 240704

Playful Minds: Humorous Peer Play and Social Understanding in Childhood

This is one of the rare studies which connects children's humour production during peer play with cognitive markers of development (receptive vocabulary, executive functions) and different indicators of theory of mind. Noteworthy is the observation of children's spontaneous humour production (in 7 categories) while playing with a peer. How the 7 observational categories of humour production were derived and what they imply should be communicated in greater depths, for example, in the body of the paper and not in a supplement.

Right now, the paper focusses on effects of children's theory of mind while excluding partner effects and markers of cognitive development. Results include a pattern of weak correlations and a multi-level-analysis confirming these associations. In my opinion, the paper would gain if the partner effects were fleshed out in more detail, because these were strong effects – much stronger, in fact, than the effects of individual children's social understanding. Beyond (social) cognition, social and situational factors seem to be very important when it comes to produce jokes, puns, or other forms of humor.

Weak points include the cross-sectional nature of the study which precludes the analysis of the direction of effects, the volunteer sample, and the lack of precision in the predictions.

I would suggest that you refocus, rewrite, and resubmit the ms.

Abstract: What is a diverse community sample? Please describe it in terms of SES.

Predictions should be included in the abstract, especially when they were registered beforehand.

Line 38: Humour is a driver of the development of socio-emotional, cognitive, and language skills^{9,10}- please look for better and more up-to-date references

Line 64: "while also driving development of new skills 29-31" Reference 31 does not support your claim. Please restate.

Line 80: "Given that research highlights that children's behaviours are often attributed to dyadic influences 80 between play partners²⁹, the extent to which child-level factors – such as social understanding skills – 81 are associated with humour production over and above partner effects is not well understood." This is the central sentence of the first 80 lines.

Please tell us what could be gained if we knew more about child factors on humour production when excluding partner effects. This should be a main message which should appear in the first paragraph.

Lines 90-96: When stated in a more technical language, you seem to make 2 predictions: (1) Children's social understanding covaries positively? concurrently with their humour production and (2) Children's social understanding covaries positively? concurrently with their humour production after partner effects and child-level covariates have been considered. If

About prediction 1: Your claim would become stronger, if you included the child-level covariates already at this step.

Why child-level covariates (such as age, gender, language skills or executive functions) may impinge upon children's humour production, should be explained in 1-2 sentences for each of these variables beforehand.

Line 157-159: There was no significant association between total observed or parent reported humor production in Table 2. Clowning, however, correlated positively with age, parent report, and receptive vocabulary. Therefore, it is important to include the word "total" in the sentence "We did not 157 detect evidence of a significant association between children's humour (observed or parent reported) 158 and child age, working memory, inhibitory control, or receptive vocabulary."

Line 165: same here: please clarify that you mean "total observed humour".

Line 170: The size of the correlations is missing here.

Line 190: That humour production in peer play has been understudied is not a good reason why it is important and why the current study makes an important contribution. I appreciate the effort that went into observational coding, and I am sure that you can find better reasons.

Line 195: Can you calculate how many children produced humour at least once? 99,7% means that 1 child did not produce any humour?

Lines 196-202: you are summarizing your results. Can you connect them this prior research? I am sure, others have studied humorous peer interactions before.

Line 243: I agree that partners play a big role when it comes to producing humour. Some partners can inspire each other to transform their relationship and the world around them by using humour whereas other partners may not join in. Would it be possible to do sequential analyses? Are the data set up in a way that it is possible to examine whether one child's humour tends to stimulate the other's humour? Perhaps that would be a different study.

In another analysis, partner A's and partner B's production of total humour can be correlated (with ICCs), to examine whether they inspire each other. The same analysis can be run with each of the other 7 subcategories (or only with those that were used more often). Can you tell whether the partner laughed at the target child's humour? Being able to set a friend up to laugh at one's joke shows a great connectedness.

Line 268: Right now, the last paragraph is a bit defensive. It can be strengthened.

Procedure

Line 291: how long did the Zoom session last? How much were the children required to use the mouse? According to our experience during this time, most 5- to 7- year-olds had difficulties operating a computer mouse and tended to get tired quickly.

Line 332: You established a good interrater reliability overall. Was it also good for all your 7 codes? Please include the kappa values for each of your codes.

Line 347: Social-cognitive tasks please include for each of the following measures a study that demonstrates the validity of the task.

Line 393: For all child covariates: range of scores and alphas are missing.

Line 550: Please include the maximum and minimum number of children in Table 2

Line 552: Please include the number of children in Table 3

Version 1:

Decision Letter:

Dear Dr Paine,

Your manuscript titled "Playful Minds: Humorous Peer Play and Social Understanding in Childhood" has now been seen by our reviewers, whose comments appear below. In light of their advice I am delighted to say that we are happy, in principle, to publish a suitably revised version in Communications Psychology.

We therefore invite you to revise your paper one last time to address the remaining concerns of our reviewers and a list of editorial requests. At the same time we ask that you edit your manuscript to comply with our format requirements and to maximise the accessibility and therefore the impact of your work.

EDITORIAL REQUESTS:

Please do not remove any preregistered analyses from the manuscript. All originally preregistered hypotheses and analyses should be included, unless scientifically unsound, in which case the deviation needs to be highlighted and explained. The full policy is here: <https://www.nature.com/commspsychol/editorial-policies/preregistration-policy>. We appreciate the request to report reliabilities for the study measurements, however, we do not consider it necessary this be done using Cronbach's alpha. You may wish to consider other indicators (cf. McNeish, 2018, <https://doi.org/10.1037/met0000144>). Low reliabilities should be discussed in the Limitations section.

SUBMISSION INFORMATION:

OPEN ACCESS:

* **DATA AVAILABILITY:**

Link Redacted

Best regards,

Jennifer Bellingtier

Jennifer Bellingtier, PhD
Senior Editor
Communications Psychology

REVIEWERS' EXPERTISE:

Reviewer #1 Social-cognitive development
Reviewer #2 Social-cognitive development

REVIEWERS' COMMENTS:

Reviewer #1 (Remarks to the Author):

I'm satisfied with the revised manuscript and have no further comments.

Reviewer #2 (Remarks to the Author):

Dear authors,

Thanks for updating the ms. in the proposed direction. I am still impressed by your observational coding, and I appreciate that humour can be a window into children's social and emotional understanding.

Most of my questions have been answered in a satisfactory way. There remain, however, some "fine points" in terms of

methods:

- It is indeed notable that you achieved an overall kappa of 0.82. The question remains, however, whether each category of humour was individually reliable across observers. Please add the kappas for each category
- For the silent film, the internal consistency is rather low ($\alpha = .52$). Can you improve internal consistency by using only some films (and leaving out the others)? Otherwise the measure needs to be deleted.
- For retrodictive mindreading, it is a bit higher, but with $\alpha = .59$ it is still below .60 which is acceptable. If not fixed, this would also call for deleting this measure.
- Alphas as measures of internal consistency are also needed for the child covariates.
- The low alphas should be mentioned in the limitations.

Point-by-point response to referees' comments

Thank you for inviting us to revise and resubmit our manuscript, '**Playful Minds: Humorous Peer Play and Social Understanding in Childhood**'. We would like to thank both reviewers for their positive feedback and constructive comments. In view of the comments from the reviewers and from the editor, we have made substantial revisions to the manuscript. Here we describe the comments and our changes to the document.

Reviewer 1

We would first like to thank Reviewer 1 for their positive comments about the manuscript regarding the novelty of the study and results, and recommendation for publication.

- 1. There needs to be a better and convincing presentation of how the different tests were selected, and how they relate to the cognitive and social abilities relevant for the research question.**

We have revised the Introduction on p. 4 to include further justification for why age-appropriate measures of children's understanding of both mental (cognitive) states and emotional cues is needed in this investigation. We hope this has provided more context for our choice of social understanding tasks.

- 2. Data was collected during the Covid-19 pandemics. It would be relevant to know the authors' discussion of how zoom based data collection can affect the quality and characteristics of data?**

Thank you for this suggestion, we have added text to the Discussion about the benefits and challenges to collecting data online on p. 11.

- 3. How were the participants selected?**

This study involved a volunteer sample. We have added some more information about the recruitment procedure to p.13.

- 4. The authors can provide some review on the different kind of humor, and its relation to children's age (e.g., analysed in naturalistic interactions).**

Additional description of types of humour observed at different ages has been added on pp. 2-3.

Reviewer 2

We would also like to thank Reviewer 2 for their positive comments about the manuscript, and particularly for pointing out that studies focusing on children's humour production in peer play are rare, and that our observational approach is noteworthy.

- 1. Abstract: What is a diverse community sample? Please describe it in terms of SES.**

Thank you for highlighting that this was not clear in the manuscript. We used this term as a reflection of the sociodemographic characteristics of the sample (i.e., over half having

a gross household income of less than UK national average). To increase clarity, we have removed the term 'diverse' from the Abstract and from the Participants section. Do to ethical constraints we did not collect data on participating families' postcodes, so we are not able to determine SES rankings (e.g., using the Wales Index of Multiple Deprivation).

2. Predictions should be included in the abstract, especially when they were registered beforehand.

We have added the pre-registered prediction to the Abstract on p. 1.

3. Line 38: Humour is a driver of the development of socio-emotional, cognitive, and language skills 9,10- please look for better and more up-to-date references

Thank you for this suggestion, we cited some key theory but understand some references were a little dated. We have amended this to three more up-to-date references (see p. 2).

4. Line 64: “while also driving development of new skills 29-31” Reference 31 does not support your claim. Please restate.

We have rewritten this section of the paper, see p. 3.

5. Line 80: “Given that research highlights that children’s behaviours are often attributed to dyadic influences 80 between play partners²⁹, the extent to which child-level factors – such as social understanding skills – 81 are associated with humour production over and above partner effects is not well understood.” This is the central sentence of the first 80 lines. Please tell us what could be gained if we knew more about child factors on humour production when excluding partner effects. This should be a main message which should appear in the first paragraph.

We have developed this argument on p. 4 of the manuscript, and we have added this message in the first paragraph of the Introduction on p.2.

6. Lines 90-96: When stated in a more technical language, you seem to make 2 predictions: (1) Children’s social understanding covaries positively? concurrently with their humour production and (2) Children’s social understanding covaries positively? concurrently with their humour production after partner effects and child-level covariates have been considered. If About prediction 1: Your claim would become stronger, if you included the child-level covariates already at this step.

Child and dyad-level factors associated with children’s humour production with peers – to our knowledge – have never been studied before and there is very little research that has investigated child-level covariates of humour production and some evidence for child-level covariates is mixed (e.g., gender – more information about this has been added to respond to R1). Therefore, part of what we wanted to determine in this manuscript was which covariates we should control for, so we took an iterative approach to understanding factors associated with humour production to control in follow up analyses, rather than controlling for

all potential covariates at the first step – this was in accordance with our pre-registration. We have added some text to clarify this approach on p. 5 of the manuscript.

- 7. Why child-level covariates (such as age, gender, language skills or executive functions) may impinge upon children's humour production, should be explained in 1-2 sentences for each of these variables beforehand.**

We have added text to p.3 of the manuscript to integrate this into the review of the literature.

- 8. Line 157-159: There was no significant association between total observed or parent reported humor production in Table 2. Clowning, however, correlated positively with age, parent report, and receptive vocabulary. Therefore, it is important to include the word "total" in the sentence "We did not 157 detect evidence of a significant association between children's humour (observed or parent reported) and child age, working memory, inhibitory control, or receptive vocabulary."**

This is added to p. 8.

- 9. Line 165: same here: please clarify that you mean "total observed humour".**

This is also added on p. 8.

- 10. Line 170: The size of the correlations is missing here.**

Thank you for noticing this error, these have been added to p.8.

- 11. Line 190: That humour production in peer play has been understudied is not a good reason why it is important and why the current study makes an important contribution. I appreciate the effort that went into observational coding, and I am sure that you can find better reasons.**

We have rewritten the opening paragraph of the Discussion on p. 9.

- 12. Line 195: Can you calculate how many children produced humour at least once? 99,7% means that 1 child did not produce any humour?**

p. 7 states that all children except one produced humour at least once.

- 13. Lines 196-202: you are summarizing your results. Can you connect them this prior research? I am sure, others have studied humorous peer interactions before.**

As with point 10, we have rewritten the opening paragraph of the Discussion section, and we have added a peer play citation on p. 9.

- 14. Line 243: I agree that partners play a big role when it comes to producing humour. Some partners can inspire each other to transform their relationship and the world around them by using humour whereas other partners may not join in. Would it be possible to do sequential analyses? Are the data set up in a way that it is possible to examine whether one child's humour tends to stimulate the other's humour? Perhaps that would be a different study.**

We agree with the reviewer that this is such an interesting question, which we have reflected on in the Discussion on p.11. Our dataset is not currently set up for sequential analysis and is not directly in scope of our current paper. We are however very keen to investigate this in our next papers.

15. In another analysis, partner A's und partner B's production of total humour can be correlated (with ICCs), to examine whether they inspire each other. The same analysis can be run with each of the other 7 subcategories (or only with those that were used more often). Can you tell whether the partner laughed at the target child's humour? Being able to set a friend up to laugh at one's joke shows a great connectedness.

Thank you for this suggestion, we have run ICCs and included them for all subcategories of humour – these are included in Table 1 (p. 22) and are summarised on p. 7. As with point 13, we agree that children's responses to one another's humour is really interesting and important. Unfortunately, it is out of the scope for the main analysis that we pre-registered for this paper, but in the future we plan to conduct a more detailed analysis of children's responses to one another's humour, e.g., positive affect, imitation, extensions of humorous acts, and negative responses. We have added more discussion about this avenue for future research on p. 10-11 of the manuscript.

16. Line 268: Right now, the last paragraph is a bit defensive. It can be strengthened.

We have looked at this paragraph again and edited it (p. 12).

17. Line 291: how long did the Zoom session last? How much were the children required to use the mouse? According to our experience during this time, most 5- to 7- year-olds had difficulties operating a computer mouse and tended to get tired quickly.

The Zoom sessions lasted approximately 1-hr (p. 13). No tasks in the Zoom assessment required children to use a mouse – all assessments were presented on the screen by the experimenter. We have added text to p.13 to describe this.

18. Line 332: You established a good interrater reliability overall. Was it also good for all your 7 codes? Please include the kappa values for each of your codes.

We can confirm that the kappa reported in the manuscript was not computed as a total count of humour, or individually per code (i.e., total count of sound play). Rather, the reported kappa is the agreement across all 7 categorical humour codes, that *after* establishing reliability were summed to create a total humour code. In this way, the calculation of our reliability was as conservative as possible, and as such it is notable that we achieved a kappa of 0.82. We have added some text to p.15 to clarify this.

19. Line 347: Social-cognitive tasks please include for each of the following measures a study that demonstrates the validity of the task.

Additional citations have been added for all social understanding tasks, p. 15-16.

20. Line 393: For all child covariates: range of scores and alphas are missing.

We have added alphas for all tasks in the study where appropriate, see highlights on p. 15 and 16. We added the range of scores to the bottom of Table 2 on p. 22.

21. Line 550: Please include the maximum and minimum number of children in Table 2

This has been added on p. 22.

22. Line 552: Please include the number of children in Table 3

This has been added on p. 23.